# Sustainability of General Population Screening for Steatotic Liver Disease: A Proof-of-Concept Study

**DOI:** 10.3390/healthcare13070759

**Published:** 2025-03-28

**Authors:** Laura De Rosa, Gabriele Ricco, Maurizia Rossana Brunetto, Ferruccio Bonino, Francesco Faita

**Affiliations:** 1Institute of Clinical Physiology, National Research Council, 56124 Pisa, Italy; lauraderosa@cnr.it; 2Hepatology Unit and Laboratory of Molecular Genetics and Pathology of Hepatitis Viruses, Reference Centre of the Tuscany Region for Chronic Liver Disease and Cancer, University Hospital of Pisa (AOUP), 56124 Pisa, Italy; riccogabriele@gmail.com (G.R.); maurizia.brunetto@unipi.it (M.R.B.); 3Department of Clinical and Experimental Medicine, University of Pisa, 56126 Pisa, Italy; 4Institute of Biostructure and Bioimaging, National Research Council, Via De Amicis 95, 80145 Naples, Italy; ferruccio.bonino@unipi.it

**Keywords:** liver disease, steatotic liver disease, steato-hepatitis, liver stiffness, transient elastography, ultrasound attenuation parameter, primary care

## Abstract

Background: Steatotic liver disease (SLD) is a growing global health concern and may progress to more advanced liver diseases (i.e., fibrosis, cirrhosis, and hepatocellular carcinoma). Early identification of individuals at risk through effective screening strategies is crucial for timely intervention and management. The aim of this population-based study was to evaluate the feasibility of mean/large-scale screening and its importance by analyzing key risk factors, such as metabolic and lifestyle-related determinants. Methods: This cross-sectional study involved 387 subjects aged 18 to 89 years in a remote rural area that stretches among the valleys at the foot of the Apennines and the Apuan Alps. Anthropometric and demographic data were recorded, together with the measurement of blood pressure and cardiac rhythm. Furthermore, US-based liver stiffness (LS) and the ultrasound attenuation parameter (UAP) using the ILivTouch (Hisky Medical, Wuxi, China) device were performed. All data were analyzed with SPSS version 28. Univariate and multivariate analyses were conducted to identify significant predictors of both LS and UAP. Results: Significant associations are observed between elevated LS and UAP values and risk factors, such as BMI and waist circumference (BMI and waist with R = 0.45 and R = 0.34, R = 0.29 and R = 0.28; respectively, for UAP and LS; all with *p* < 0.001). The presence of hypertension is associated with a high value of LS (*p* < 0.05), and high UAP with alcohol consumption and sugary coffee intake habit (*p* < 0.001 and, *p* < 0.05, respectively). Conclusions: General population screening for SLD is feasible, sustainable, and useful to identify both individuals at risk and patients with progressive liver disease.

## 1. Introduction

Steatotic liver disease (SLD) affects approximately 30% of the general population in Western Countries and its prevalence is expected to increase significantly in the next decade [1]. The epidemic spread of SLD is associated not only with an increasing incidence of metabolic dysfunction-associated steatotic liver disease (MASLD) and its hepatic complications (steato-hepatitis, cirrhosis, and hepatocellular carcinoma), but also with systemic comorbidities, such as obesity, type 2 diabetes (T2D), and cardiovascular, chronic kidney, neurodegenerative, and neoplastic diseases [2]. So far, the management of patients with MASLD is based on lifestyle modifications, including personalized dietary interventions, physical activity, and weight reduction, which have demonstrated efficacy in reducing hepatic fat content and improving liver enzymes [3]. The current clinical guidelines propose algorithms aimed at identifying patients at a higher risk of progressive liver disease based on the use of non-invasive tests (NITs) [3,4]. Specifically, the Fibrosis-4 (Fib-4) index for liver fibrosis [5,6] (derived from the combination of aspartate aminotransferase, alanine aminotransferase, platelet count, and age) is recommend using a cut-off of >1.3 as indicative for a secondary risk assessment by transient elastography (TE); subsequently, patients with TE > 8 kPa should be referred to a hepatology specialist for further clinical and diagnostic evaluations [3,4]. Recently, new drugs, such as sodium/glucose cotransporter-2 inhibitors and glucagon-like peptide-1 agonists, have shown for the first time an improvement of both cardiovascular and liver steatosis outcomes in T2D patients [7,8,9,10,11]. Similarly, thyroid hormone (TH) analogs, such as resmetirom, have been approved by the FDA for the treatment of MAFLD patients with F2-F3 fibrosis [12,13,14,15,16]. In the near future, new molecules, such as the analogs of hepatokine, fibroblast growth factor 21 (FGF21), capable of modulating the metabolic profile (glucose and lipid metabolism) in an earlier stage of the disease, prior to overt type 2 diabetes (T2D), will become available [17,18,19]. All this evidence emphasizes the importance of the early diagnosis and treatment of steato-hepatitis. In particular, the identification of subjects with asymptomatic SLD and steato-hepatitis at risk of liver disease progression is essential to prevent the development of advanced liver fibrosis—strongly associated with advanced liver disease. With these aims in mind, we performed a proof-of-concept study to assess the feasibility and sustainability of a general population screening for SLD in small villages of Lunigiana, a territory in northern Tuscany that stretches among the valleys at the foot of the Apennines and the Apuan Alps.

## 2. Materials and Methods

### 2.1. Study Design and Population

We conducted a general population-based study on the inhabitants of a small geographical area of Tuscany (i.e., Lunigiana). Volunteers who presented at general healthcare facilities on 5 free screening events organized by local social and healthcare volunteer organizations [Red Cross Committee of Albiano Magra and Christian Brotherhood of Bagnone, https://misericordiabagnone.com/ (accessed on 15 February 2025)] in 3 different small villages (Albiano Magra, Bagnone, and Terrarossa Castle) located in a rural area of Lunigiana (Massa–Carrara province) at the north-western extreme of Tuscany, Italy, were included in this study. All subjects provided informed written consent. Overall, we screened 387 individuals with a mean age of 63 ± 14 years (range: 18–89 years) [193 males, 62 ± 15 (range: 18–89) years, and 194 females, mean age 63 ± 12 (range: 28–87) years].

Demographic data (sex, age, and occupation), a short anamnesis (lifestyle and disease history), and blood pressure, height, weight, body mass index (BMI), and waist circumference were registered by nurse volunteers for all subjects. All of them were asked to answer a general questionnaire about their lifestyle, e.g., physical activity levels, coffee intake, and alcohol consumption.

Furthermore, their UAP and liver stiffness (LS) analyses were performed by volunteer medical doctors from the Hepatology Unit and Reference Centre of the Tuscany Region for Chronic Liver Disease and Cancer of the University Hospital of Pisa (AOUP), Pisa, Italy. The ultrasound (US)-based diagnostic system for UAP and LS measurements was ILivTouch (Hisky Medical, Wuxi, China) and all tests were performed on the right liver lobe after >3 h of fasting through intercostal spaces with the subjects lying on their backs with the right arm in the maximal abduction position. A US guide was used to identify a target liver area at least 6 cm thick without major vascular structures or the gallbladder. The procedure was based on at least 10 repeated measurements and was considered valid with the interquartile range/median values of the measurements (IQR/M) ≤ 30%. LS was recorded in kilopascals and liver fat content as an ultrasound attenuation parameter (UAP) (dB/m); median values of all measurements were considered. For each pts IQR of LS, and the success rate were recorded.

This study complies with the STROBE reporting guidelines [20].

### 2.2. Statistical Analysis

Descriptive statistics were assessed for all variables. The categorical data are presented as counts and percentages, while the mean and standard deviation are reported for continuous normally distributed data. The Student’s *T*-test or chi-squared test was used to compare all variables stratified within sex groups and also by stratifying variables by LS and UAP values (i.e., LS </≥ 7 kPa and UAP </≥ 230 dB/m) for continuous and categorical variables, respectively. Pearson’s/Spearman’s correlation was assessed to analyze the association with demographic clinic–pathologic data, overall and by splitting the population depending on sex and presence/absence of T2D. In addition, Student’s *T*-test was performed to evaluate the differences between LS and UAP depending on the presence of T2D. Univariate and multivariate linear regression models were also used in order to identify predictors associated with LS and UAP. Correlations were considered significant with R ≥ 0.300 and *p* < 0.05. All statistics were performed with SPSS Statistics version 28 (IBM, Armonk, NY, USA).

## 3. Results

The characteristics of the study population are presented in Table 1, overall and after stratifying by sex.

### 3.1. Demographic, Physical, and Lifestyle Characteristics

As far as the birthplace of the volunteers was concerned, 198 of them (51%) were born in the Massa–Carrara province, while 80 (21%) were born in the nearby La Spezia province; 91 (23%) were native Italians from other areas, and 18 (5%) were immigrants. According to their occupation, most of subjects (37%) were retired, and only 1% unemployed.

Overall, BMI was 26.2 ± 4.4 kg/m^2^ and the waist circumference was 98 ± 15 cm. Regarding lifestyle habits, the majority of the population (70%) did not smoke and a further 16% reported having smoked in the past, while 12% currently smoke. About 2% of the sample did not provide information for this question. As for alcohol consumption, 43% of subjects did not consume alcohol at all, while 54% of them reported consuming alcohol. Among these, 20% drank occasionally, 18% drank daily, 13% rarely consumed alcohol, and 49% did not provide information about drinking. Finally, only a small percentage (3%) did not respond to the alcohol consumption question. Moreover, 53% of the volunteers had coffee with sugar, whereas 37% had it without sugar. A smaller group (7%) stated that they used to drink sweetened coffee in the past. The data about coffee consumption information were not available for 3% of the population.

Looking at the physical activity levels, 52% of the enrolled subjects did not disclose their physical activity levels. Among the remaining respondents, 23% declared to have moderate physical activity, 13% maintain good physical activity levels, and 12% reported insufficient physical activity.

BMI, waist circumference, and alcohol consumption habits showed statistically significant differences when comparing female/male subgroups (higher in the male group, *p* < 0.001 for all variables) (Table 1: last column represents the *p*-value of the *T*-test/chi-squared test of all variables comparing them by sex).

### 3.2. Disease History

Out of all the participants, 33% had arterial hypertension (HT), 5% were affected by type 2 diabetes (T2D), 6% of them reported to have cardiovascular disease, and approximately 1% had a digestive disease. Regarding drugs usage, the majority of the population (62%) is on pharmacological treatment, while 11% denies any use of drugs. Information about pharmacological treatment was not available for 27% of the subjects.

### 3.3. Measured Parameters

US-based examinations showed UAP values of 256 ± 44 dB/m and an LS measurement equal to 7.5 ± 37 kPa. Overall, systolic and diastolic pressure values were 137 ± 18 mmHg and 77 ± 10 mmHg, respectively, with a heart rate of 75 ± 13 bpm and an oxygen saturation percentage of 97 ± 2%. Statistically, no significant differences were found in the UAP and LS values between males and females.

### 3.4. Correlations Between UAP and LS and Other Characteristics

Correlation analysis showed significant (*p* < 0.001) correlations between UAP and both BMI and waist circumference (R = 0.45 and R = 0.34, respectively), whereas LS was significantly correlated only with weight (R = 0.30, *p* < 0.001). Interestingly, the correlations of UAP versus BMI and waist circumference were stronger in males (R = 0.48 and R = 0.40 respectively, with *p* < 0.001 for both) compared to females (R = 0.38, *p* < 0.001 and R = 0.20, *p* = 0.006, respectively). Limiting the analysis to the subpopulation affected by T2D, BMI was strongly correlated with UAP (R = 0.55, *p* = 0.03) compared to the non-diabetic subjects (R = 0.44, *p* < 0.001). It is noteworthy that, no significant correlations were found between the consumption of sweetened coffee or with BMI or waist circumference versus LS or UAP measurements. In Table 2 and Table 3, the univariate and multivariate models’ results are shown for UAP and LS measurements, respectively.

As reported in Table 2, in the univariate analysis, BMI, and waist circumference values are found to have significant positive associations (*p* < 0.001), while sex shows a significant negative relationship (*p* < 0.001) with UAP. Other variables, including age, hypertension (HT), type 2 diabetes (T2D), alcohol consumption, smoking status, coffee consumption (sugary, non-sugary, and ex-sugary), and physical activity levels, did not show significant associations individually.

In the multivariate analysis, two models were evaluated, and, in both models, BMI remained significant (*p* < 0.001), while other variables that were significant following the univariate analysis lost their significance.

Furthermore, the univariate analysis in Table 3 shows that BMI (*p* < 0.001) and waist circumference (*p* < 0.001) have significant positive associations, while sex demonstrates a significant negative association (*p* = 0.004) with LS. All other variables, including hypertension (HT); type 2 diabetes (T2D); alcohol consumption; smoking status (never, ex, and current); sugary coffee, non-sugary coffee, and ex-sugary coffee drinkers; and physical activity (poor, moderate, and good), showed no significant associations with stiffness variable.

In the multivariate analysis, as reported for UAP, even for LS, the only variable that remained significant was BMI (*p* < 0.05) for both models.

The effect of the presence of T2D on both LS and UAP values was also analyzed, and the results are reported in Table 4.

Both UAP and LS showed significant correlations with BMI (R = 0.441, R = 0.334, for UAP and LS, respectively) and waist circumference (R = 0.324, R = 0.307, for UAP and LS, respectively) in the group without T2D (all with *p* < 0.001). Interesting, correlations between UAP and both BMI and waist circumference became stronger in the subgroup of subjects with T2D (N = 18) (R = 0.553/0.468 *p* < 0.05 for both, respectively); on the other hand, no significant correlation was found with LS. Furthermore, by comparing LS and CAP measurements in the group of subjects with/without T2D stratified by sex, statistical differences were found only for LS only in the female group (*p* < 0.05).

In addition, statistical differences were found between lifestyle habits, disease prevalence, and anthropometric data by the splitting population according to LS (</≥7 kPa) and UAP (</≥230 dB/m). These results are reported in Table 5.

## 4. Discussion

The growing healthcare burden posed by the increasing worldwide prevalence of Steatotic Liver Disease (SLD) and the availability of non-invasive devices/tools for the diagnosis of steatosis and fibrosis [21,22,23,24,25] prompts studies on the cost-effectiveness of screening the general population for SLD. In a proof-of-concept study, we tested whether such a screening was feasible and sustainable, even in a remote rural area that stretches among the valleys at the foot of the Apennines, and the Apuan Alps provided the necessary contribution of the volunteers of local social and healthcare organizations coordinated with general practitioners. Each subject spent about 20 min for both the registration and visit with half time for the UAP and LS examinations. Overall, the prevalence of fatty liver (UAP ≥ 230 dB/m) and liver stiffness (LS ≥ 7 kPa, used to identify the risk of ongoing fibro-inflammatory liver pathology) [26] was 72% (76.7% in males and 67.5% in females) and 40% (44.6% in males and 35.6% in females), respectively. Both values are significantly higher than those reported in the largest screening study (5.7 million participants) recently performed in China [27] using the same type of equipment and aimed to estimate the prevalence of various grades of liver steatosis and fibrosis assessed by LS in the general population. This difference stems firstly from the inevitable recruitment bias due to the communication/information campaign aimed at SLD and associated pathologies, and secondly to the high median age of the study population. In fact, in Italy the current prevalence of SLD in the general population is reported to correlate significantly with age; the age-related prevalence shows increasing values corresponding to the decade of age, namely 10% at 10 years of age till 60% at 60 years [28]. Furthermore, the high prevalence of SLD stems from other characteristics of our study population, such as sedentary lifestyle: only 13% (16% males, 10% females) declared having good and 23% (21% males, 25% females) moderate physical activity. Overall, only 12% admitted smoking, but 54% alcohol intake (daily in 18%) and 53% the consumption of sugar with coffee that can be considered a reliable stigma for sugar craving. In keeping with these results, we found a relatively high prevalence of comorbidities typically associated with SLD: overall 18.5% had T2D, 33% HT, 24.6% cardiovascular disease other than AH (CVD), 21% other chronic diseases, and overall 62% were taking drugs for chronic diseases. Following the univariate analysis, both UAP and LS correlated significantly with sex, BMI, and waist circumference (Table 2 and Table 3, for UAP and LS, respectively), and in addition, UAP correlated with physical activity, but following the multivariate analysis, the BMI remains the only variable significantly correlated with both UAP and LS. Therefore, BMI emerged as the most consistent and significant predictor for both UAP and LS variables across models tested, while other factors, including sex and waist circumference, were significant only in the univariate analysis but lost significance when adjusted for other variables.

As far as lifestyle is concerned, UAP > 230 dB/m, but not LS > 7 kPa, was associated with the assumption of sugar with coffee (*p* < 0.05) and alcohol (*p* < 0.001). Interestingly, in patients with T2D, only UAP but not LS was highly correlated with BMI, whereas overall, as well as in non-T2D subjects, both UAP and LS correlated with BMI.

These findings are in agreement with many studies on humans and animal models showing that SLD is a pathologic condition underlying the risk of T2D and other metabolic disorders [29]. The results of this small study confirm that the combined measure of UAP and LS with the currently available automated US instruments provide a reliable screening method for the early identification of SLD subjects, and among them, of patients at risk of progressive liver disease because of ongoing liver necro-inflammation (steato-hepatitis, SH). In light of the newly available treatments, an earlier SH diagnosis provides a the timelier treatment of patients before the progression of liver fibrosis, and in patients with T2D and CVD highlights the risk of systemic disease progression and complications. Our population-study was based on a cohort of volunteers from a small area of Tuscany (Italy), and this posed a major limitation caused by the selection bias. Nevertheless, this proof-of-concept study underlines the validity of the combined measure of the ultrasound attenuation parameter and liver stiffness provided by transient-elastography-based devices as an effective and sustainable non-invasive screening tool for liver steatosis and early liver disease diagnosis. The results hold true independently from the inability of LS to distinguish the relative impact of the two major components of liver damage, namely inflammation and fibrosis. Future studies are necessary on larger cohorts aiming to identify lower TE cut-offs for the early identification of steato-hepatitis since its earlier diagnosis would allow lifestyle modifications and/or pharmacological interventions with the highest chance of preventing liver and systemic disease progression.

## 5. Conclusions

This proof-of-concept study in a broad context highlights the importance of screening the general population for SLD and using appropriate and well-defined transient elastography cut-offs for the early identification of patients with steato-hepatitis, and referring them for timely therapy that is essential to prevent the development of irreversible metabolic diseases and liver-related complications.

## Figures and Tables

**Table 1 healthcare-13-00759-t001:** Description of the study population overall and split by sex. *p*-value indicates the *T*-test/chi-squared test results for different sex variables.

Variable	Overall N = 387	Male N = 193	Female N = 194	*p*-Value
Age (years)	63 ± 14	62 ± 15	63 ± 12	n.s.
BMI (kg/m^2^)	26.2 ± 4.4	27.1 ± 4.3	25.4 ± 4.4	<0.001
Waist circumference (cm)	98 ± 15	103 ± 15	93 ± 15	<0.001
HT (yes)	128 (33%)	56	72	n.s.
T2D (yes)	18 (5%)	8	10	n.s.
Alcohol consumption (yes:no)	210:167 (54:43)%	121:65	89:102	<0.001
Smoker (yes:no:ex)	45:273:63 (12:70:16)%	22:127:39	23:146:24	n.s.
Sugary coffee (yes:no:ex)	204:145:27 (53:37:7)%	105:66:13	99:79:14	n.s.
Physical activity (Lacking:Moderate:Good)	48:89:49 (12:23:13)%	26:40:30	22:49:19	n.s.
UAP (dB/m)	256 ± 44	264 ± 46	249 ± 42	n.s.
LS (kPa)	7.5 ± 3.7	8.0 ± 4.3	138 ± 21	n.s.
Systolic pressure (mmHg)	137 ± 18	135 ± 15	138 ± 21	n.s.
Diastolic pressure (mmHg)	77 ± 10	77 ± 10	76 ± 11	n.s.
HR (bpm)	75 ± 13	73 ± 13	77 ± 12	n.s.
Saturation (%)	97 ± 2	97 ± 2	97 ± 2	n.s.

BMI, body mass index; HT, arterial hypertension; T2D, type two diabetes; UAP, ultrasound attenuation parameter; LS, liver stiffness; HR, heart rate; n.s.: non-significant.

**Table 2 healthcare-13-00759-t002:** Results of the univariate and multivariate analyses for assessing the association between UAP values and all collected variables on the whole study population. Statistically significant results (*p* < 0.05) are highlighted in bold.

	Univariate		Multivariate Model 1 ^1^		Multivariate Model 2 ^1^	
Variable	R	*p*	R	*p*	R	*p*
Age	−0.102	0.534				
Sex	−15.004	**<0.001**	−9.571	0.061	−9.633	0.069
BMI	4.512	**<0.001**	3.509	**<0.001**	3.413	**<0.001**
Waist circumference	0.980	**<0.001**	0.211	0.243	0.238	0.204
HT	6.372	0.142				
T2D	18.292	0.090			6.409	0.615
Non-smokers	−1.240	0.807				
Ex-smokers	−6.463	0.293				
Smokers	10.982	0.120				
Alcohol	−7.031	0.129				
Sugary coffee	0.389	0.933				
No sugary coffee	3.924	0.409				
Ex-sugary coffee drinkers	−15.396	0.085			−10.202	0.450
Physical activity: lacking	13.476	**0.035**	8.440	0.137	7.669	0.192
Physical activity: moderate	−7.647	0.173				
Physical activity: good	−3.463	0.587				

^1^ Model 1: includes as independent predictors the variables showing a *p* < 0.05 for the univariate analysis; Model 2: like Model 1, but only variables with *p* < 0.1 for the univariate analysis are included. BMI, body mass index; HT, arterial hypertension; T2D, type two diabetes.

**Table 3 healthcare-13-00759-t003:** Univariate and multivariate linear regression analyses of LS values on the whole study population. Statistically significant results (*p* < 0.05) are highlighted in bold.

	Univariate		Multivariate Model 1 ^1^		Multivariate Model 2 ^1^	
Variable	R	*p*	R	*p*	R	*p*
Age	0.022	0.100			0.027	0.057
Sex	−1.088	**0.004**	−0.545	0.165	−0.607	0.122
BMI	0.246	**<0.001**	0.160	**0.006**	0.182	**0.002**
Waist circumference	0.068	**<0.001**	0.031	0.074	0.024	0.171
HT	0.329	0.362				
T2D	0.764	0.39				
Non-smokers	0.087	0.836				
Ex-smokers	0.030	0.954				
Smokers	−0.209	0.722				
Alcohol	0.253	0.510				
Sugary coffee	0.234	0.526				
No sugary coffee	−0.397	0.293				
Ex-sugary coffee drinkers	0.540	0.449				
Physical activity: lacking	0.814	0.200				
Physical activity: moderate	−0.588	0.290				
Physical activity: good	−0.047	0.941				

^1^ Model 1: includes as independent predictors the variables showing a *p* < 0.05 for the univariate analysis; Model 2: like Model 1, but only variables with *p* < 0.1 for the univariate analysis are included. BMI, body mass index; HT, arterial hypertension; T2D, type two diabetes.

**Table 4 healthcare-13-00759-t004:** Correlation analysis between LS and UAP versus BMI and waist circumference depending on the presence of T2D.

Variable		LS		UAP	
		R	*p*	R	*p*
BMI (kg/m^2^)	No T2D	0.334	<0.001	0.441	<0.001
	T2D	0.090	n.s.	0.553	<0.05
Waist circumference (cm)	No T2D	0.307	<0.001	0.324	<0.001
	T2D	−0.042	n.s.	0.468	<0.05

BMI, body mass index; T2D, type two diabetes; UAP, ultrasound attenuation parameter; LS, liver stiffness; n.s.: non-significant.

**Table 5 healthcare-13-00759-t005:** Analysis of differences for lifestyle markers, disease prevalence (T2D and HT), and anthropometric data between subjects with LS values higher and lower than 7 kPa and UAP </≥ 230 dB/m. Results of the *T*-test and chi-squared are reported for continuous and categorical variables.

	LS </≥ 7 kPa N = 232/155	UAP </≥ 230 dB/m N = 108/279
T2D	-	-
HT	<0.05	-
Sex	<0.05	<0.05
Alcohol	-	<0.001
Smokers	-	-
Sugary coffee	-	<0.05
Physical activity	<0.05	-
Age	<0.05	-
BMI	<0.001	<0.001
Waist circumference	<0.001	<0.01

LS, liver stiffness; UAP, ultrasound attenuation parameter; BMI, body mass index; HT, arterial hypertension; T2D, type two diabetes.

## Data Availability

The data presented in this study are available on request from the corresponding authors. Data are not publicly available due to privacy.

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
