# Peer review of "Sustainability of General Population Screening for Steatotic Liver Disease: A Proof-of-Concept Study"

_healthcare, 2025, doi:10.3390/healthcare13070759_

Round 1
Reviewer 1 Report
Comments and Suggestions for Authors
- There are lot of study reported on non-invasive biomarker to identify or monitor progression of liver disease using liver stiffness (LS) and ultrasound attenuation parameter (UAP) data with basic anthropometric and demographic data. The only different I can see it that the study population was part of Italy. The authors did not mention clearly how these data will help them to identify it.
- The authors have discussed with very limited references. I would suggest the authors include few more contexts with references in the discussion section.
- The authors did not mention the limitations of the study, drawback and future perspective of the study in the population. So, it would be good if the authors could mention those parts wither in end of the discussion or conclusion section.
- The authors did not mention the details about Institutional Review Board to conduct studies on humans.
- Mention what is FIB-4 in the page 1
- In pg. 2 of 8, line 47, it should be “new drugs such as……”. Please check typographical error.
- In page 2 of 8, rephrase the lines 54-58 starting with “All these……. advanced liver disease” to make easily understandable.
- I would suggest the authors to include exclusion and inclusion criteria for choosing the cohorts for the study in the materials and method section.
- In table 1, please change “not” to “no” like yes:no:ex instead of yes:not:ex
- In table 2 and 3, p value for sex, BMI, and waist circumstances were mentioned as 0.001 and 0.004. I would suggest the authors to check whether these values were p<0.001 or 0.004 or equal to these values.
- In general, check all the statistical data and typographical errors if any throughout the manuscript.
Few sentences should be rephrased and typographical errors should be checked throughout the manuscript:
- In pg. 2 of 8, line 47, it should be “new drugs such as……”. Please check typographical error.
- In page 2 of 8, rephrase the lines 54-58 starting with “All these……. advanced liver disease” to make easily understandable
Author Response
Comments 1: There are lot of study reported on non-invasive biomarker to identify or monitor progression of liver disease using liver stiffness (LS) and ultrasound attenuation parameter (UAP) data with basic anthropometric and demographic data. The only different I can see it that the study population was part of Italy. The authors did not mention clearly how these data will help them to identify it.
Response 1: Actually, the aim of the work was not the one commonly and extensively addressed by many studies, as underlined by the reviewer, but the sustainability and feasibility of using the LS measure as a screening tool in the general population. We have more clearly stated and underlined this in the revised version of our manuscript.
Comments 2: The authors have discussed with very limited references. I would suggest the authors include few more contexts with references in the discussion section.
Response 2: According to the Reviewer suggestion, we included 5 more papers conducting screening studies on different population worldwide, as also suggested by Reviewer 2. We added the following sentences at the beginning of the Discussion section (page 7, lines 231-235):
“The growing health care burden posed by the increasing worldwide prevalence of Steatotic Liver Disease (SLD) and the availability of non-invasive devices/tools for diagnosis of steatosis and fibrosis [21-25] prompt studies on the cost-effectiveness of screening general population for SLD. In a proof-of-concept study we tested whether such a screening was feasible and sustainable…”
Comments 3: The authors did not mention the limitations of the study, drawback and future perspective of the study in the population. So, it would be good if the authors could mention those parts wither in end of the discussion or conclusion section.
Response 3: Thank you for your very helpful comment. We agree that it is very important to discuss these aspects. Accordingly, we added some limitations of the study and possible future perspectives. Please, find below the additional period we included in the text, at the end of the Discussion section (page 8, lines 282-293):
“Our population-study was based on a cohort of volunteers of a small area of Tuscany (Italy) and this poses a major limitation caused by the selection bias. Nevertheless, this proof-of-concept study underlines the validity of the combined measure of ultrasound attenuation parameter and liver stiffness provide by transient-elastography based devices as an effective and sustainable non-invasive screening tool for liver steatosis and early liver disease diagnosis. The results hold true independently from the inability of LS to distinguish the relative impact of the two major components of liver damage, namely inflammation and fibrosis. Future studies are necessary on larger cohorts aiming to identify lower TE cut-offs for the early identification of steato-hepatitis since its earlier diagnosis would make life style modifications and/or pharmacological intervention with the highest chance of preventing liver and systemic disease progression.”
Comments 4: The authors did not mention the details about Institutional Review Board to conduct studies on humans
Response 4: Detailed information about IRB has been then added with the support of an Assistant Editor in the section Institutional Review Board Statement (page 8, lines 305-309), as follow:
“Institutional Review Board Statement: the study was conducted in accordance with the Declaration of Helsinki; as this study is based on data collected from a voluntary, non-interventional public health screening program, Ethics Committee approval was deemed unnecessary, as per Italian law TULPS R.D. 18 June 1931, n. 773 and subsequent changes/additions, since no additional interventions were performed for this study.”
Comments 5: Mention what is FIB-4 in the page 1
Response 5: Thank you very much for your suggestion. Accordingly, we added the full name of the score (non-abbreviated form) and also included two references after the first appearance of Fib-4 (page 1, lines 45-47):
- Kariyama K, Nouso K, Toyoda H, Tada T, Hiraoka A, Tsuji K, Itobayashi E, Ishikawa T, Wakuta A, Oonishi A, Kumada T, Kudo M, Group OBOTRPEFHRS, Group H. Utility of FIB4-T as a Prognostic Factor for Hepatocellular Carcinoma. Cancers (Basel). 2019 Feb 10;11(2):203. doi: 10.3390/cancers11020203. PMID: 30744175; PMCID: PMC6406758.
- Salomone F, Micek A, Godos J. Simple Scores of Fibrosis and Mortality in Patients with NAFLD: A Systematic Review with Meta-Analysis. J Clin Med. 2018 Aug 15;7(8):219. doi: 10.3390/jcm7080219. PMID: 30111756; PMCID: PMC6111765.
Comments 6: In pg. 2 of 8, line 47, it should be “new drugs such as……”. Please check typographical error.
Response 6: Thank you for pointing out this mistake. We are sorry about some typographical errors, hoping that now no more typos are left.
Comments 7: In page 2 of 8, rephrase the lines 54-58 starting with “All these……. advanced liver disease” to make easily understandable.
Response 7: Thank you, we rephrased these sentences, lines 58-61, as reported below:
“All these evidences emphasize the importance of early diagnosis and treatment of steato-hepatitis. In particular, the identification of subjects with asymptomatic SLD and steato-hepatitis at risk of liver disease progression is essential to prevent the development of advanced liver fibrosis – strongly associated with advanced liver disease.”
Comments 8: I would suggest the authors to include exclusion and inclusion criteria for choosing the cohorts for the study in the materials and method section.
Response 8: We did not apply particular inclusion/exclusion criteria for the selection of the cohort. As reported in the Material section, we included all volunteers who came during the 5 free liver screening days organized in a rural area of Tuscany (Lunigiana) in Italy since it was a pilot study of feasibility and sustainability of population screening, and any selection made would have added biases to the study. However, we added some more details about the study design at the beginning of the Materials and Methods section to make it clearer to the reader.
Comments 9: In table 1, please change “not” to “no” like yes:no:ex instead of yes:not:ex
Response 9: Thank you. Accordingly, we updated all the instances in the tables as suggested.
Comments 10: In table 2 and 3, p value for sex, BMI, and waist circumstances were mentioned as 0.001 and 0.004. I would suggest the authors to check whether these values were p<0.001 or 0.004 or equal to these values.
Response 10: We have checked all results of the univariate and multivariate analyses on both UAP and LS values. Accordingly, we found few missing of “<” where a p-value of 0.001 was reported. Now we confirm that all p-values are exactly the values determined by statistics.
Comments 11: In general, check all the statistical data and typographical errors if any throughout the manuscript.
Response 11: We checked data into Tables and into the main text and adjusted some errors.
Response to Comments on the Quality of English Language:
Point 1: In pg. 2 of 8, line 47, it should be “new drugs such as……”. Please check typographical error.
Response 1: As above reported, we fixed it.
Point 2: In page 2 of 8, rephrase the lines 54-58 starting with “All these……. advanced liver disease” to make easily understandable
Response 2: We rephrased the sentence, hoping now it is easier and smoother to read and understand.

Reviewer 2 Report
Comments and Suggestions for Authors
It is an observational study with its inherent issues
The following are suggested to enhance the manuscript:
- STROBE guidelines to be followed and the same be mentioned in the methods section
- The discussion section is inadequate as the relevant references of other data have not been discussed. The following may be examined and referenced:
a. Pandyarajan V, Gish RG, Alkhouri N, Noureddin M. Screening for Nonalcoholic Fatty Liver Disease in the Primary Care Clinic. Gastroenterol Hepatol (N Y). 2019 Jul;15(7):357-365. PMID: 31391806; PMCID: PMC6676352.
b. Dietrich CG, Rau M, Geier A. Screening for nonalcoholic fatty liver disease-when, who and how? World J Gastroenterol. 2021 Sep 21;27(35):5803-5821. doi: 10.3748/wjg.v27.i35.5803. PMID: 34629804; PMCID: PMC8475001.
c. Mohammadi T, Mohammadi B. Screening the General Population for Non-Alcoholic Fatty Liver Disease: Model Development and Validation. Arch Med Res. 2024 Apr;55(3):102987. doi: 10.1016/j.arcmed.2024.102987. Epub 2024 Mar 21. PMID: 38518527.
d. Abeysekera KWM, Fernandes GS, Hammerton G, Portal AJ, Gordon FH, Heron J, Hickman M. Prevalence of steatosis and fibrosis in young adults in the UK: a population-based study. Lancet Gastroenterol Hepatol. 2020 Mar;5(3):295-305. doi: 10.1016/S2468-1253(19)30419-4. Epub 2020 Jan 15. PMID: 31954687; PMCID: PMC7026693.
e. Zhang S, Mak LY, Yuen MF, Seto WK. Screening strategy for non-alcoholic fatty liver disease. Clin Mol Hepatol. 2023 Feb;29(Suppl):S103-S122. doi: 10.3350/cmh.2022.0336. Epub 2022 Nov 30. PMID: 36447420; PMCID: PMC10029948.
4. The limitations of the study have not been mentioned
5. The way forward should be mentioned in the concluding paragraph
Author Response
Comments 1: STROBE guidelines to be followed and the same be mentioned in the methods section
Response 1: Thank you for your very helpful suggestions. Accordingly, we added the reference to the STROBE checklist including it at the end of Study Design and Population subsection of Materials and Methods (page 3, line 105):
“This study complies with STROBE reporting guidelines [20].”
Including the following reference at lines 390-391:
[20] Cuschieri S. The STROBE guidelines. Saudi J Anaesth. 2019 Apr;13(Suppl 1):S31-S34. doi: 10.4103/sja.SJA_543_18. PMID: 30930717; PMCID: PMC6398292.
Comments 2: The discussion section is inadequate as the relevant references of other data have not been discussed. The following may be examined and referenced:
- Pandyarajan V, Gish RG, Alkhouri N, Noureddin M. Screening for Nonalcoholic Fatty Liver Disease in the Primary Care Clinic. Gastroenterol Hepatol (N Y). 2019 Jul;15(7):357-365. PMID: 31391806; PMCID: PMC6676352.
- Dietrich CG, Rau M, Geier A. Screening for nonalcoholic fatty liver disease-when, who and how? World J Gastroenterol. 2021 Sep 21;27(35):5803-5821. doi: 10.3748/wjg.v27.i35.5803. PMID: 34629804; PMCID: PMC8475001.
- Mohammadi T, Mohammadi B. Screening the General Population for Non-Alcoholic Fatty Liver Disease: Model Development and Validation. Arch Med Res. 2024 Apr;55(3):102987. doi: 10.1016/j.arcmed.2024.102987. Epub 2024 Mar 21. PMID: 38518527.
- Abeysekera KWM, Fernandes GS, Hammerton G, Portal AJ, Gordon FH, Heron J, Hickman M. Prevalence of steatosis and fibrosis in young adults in the UK: a population-based study. Lancet Gastroenterol Hepatol. 2020 Mar;5(3):295-305. doi: 10.1016/S2468-1253(19)30419-4. Epub 2020 Jan 15. PMID: 31954687; PMCID: PMC7026693.
- Zhang S, Mak LY, Yuen MF, Seto WK. Screening strategy for non-alcoholic fatty liver disease. Clin Mol Hepatol. 2023 Feb;29(Suppl):S103-S122. doi: 10.3350/cmh.2022.0336. Epub 2022 Nov 30. PMID: 36447420; PMCID: PMC10029948.
Response 2: Thank you for your comment and the references you provided. We added them in the Discussion section (page 7, lines 231-235).
The following sentence was included:
“The growing health care burden posed by the increasing worldwide prevalence of Steatotic Liver Disease (SLD) and the availability of non-invasive devices/tools for diagnosis of steatosis and fibrosis [21-25] prompt studies on the cost-effectiveness of screening general population for SLD. In a proof-of-concept study we tested whether such a screening was feasible and sustainable…”
Comments 4: The limitations of the study have not been mentioned
Response 4: As per Reviewer 1, please find below limitations of the study reported in the Discussion section (page 8, lines 282-289):
“Our population-study was based on a cohort of volunteers of a small area of Tuscany (Italy) and this poses a major limitation caused by the selection bias. Nevertheless, this proof-of-concept study underlines the validity of the combined measure of ultrasound attenuation parameter and liver stiffness provide by transient-elastography based devices as an effective and sustainable non-invasive screening tool for liver steatosis and early liver disease diagnosis. The results hold true independently from the inability of LS to distinguish the relative impact of the two major components of liver damage, namely inflammation and fibrosis.”
Comments 5: The way forward should be mentioned in the concluding paragraph
Response 5: We strongly agree about to discuss these aspects. Accordingly, possible future perspectives have been included. Please, find below the additional period we included in the text, at the end of the Discussion section (page 8, lines 289-293):
”Future studies are necessary on larger cohorts aiming to identify lower TE cut-offs for the early identification of steato-hepatitis since its earlier diagnosis would make life style modifications and/or pharmacological intervention with the highest chance of preventing liver and systemic disease progression.”

Reviewer 3 Report
Comments and Suggestions for Authors
The authors have done a really interesting research and the overall quality of the article is impressive. I am in the favor of the publication and believe that early detection could definitely change the overall outcome in this most commonly encountered liver disease. It is interesting how the authors coordinate the research. The use of sugary coffee was interesting and why was the overall sugar intake not counted and only the sugar in the coffee was given importance? I also suggest the authors to expand their research in future and include more subjects.
I have some minor suggestions:
In Line 36: The statement, 'The epidemic spread of SLD associates not only with an increasing incidence……' needs a change.
Suggestion: The epidemic spread of SLD is associated not only with ……
In line 37: MASLD should be Metabolic dysfunction-associated steatotic liver disease.
In 121: 'statistically differences' should be 'stastically significant differences'.
Line 137: 'oxigen' should be 'oxygen'
Comments on the Quality of English LanguageThe English language use is impressive. The minor suggestions I have pointed out above should improve the overall quality.
Author Response
Comments 1: The authors have done a really interesting research and the overall quality of the article is impressive. I am in the favor of the publication and believe that early detection could definitely change the overall outcome in this most commonly encountered liver disease. It is interesting how the authors coordinate the research. The use of sugary coffee was interesting and why was the overall sugar intake not counted and only the sugar in the coffee was given importance? I also suggest the authors to expand their research in future and include more subjects.
Response 1: Thank you to the Reviewer comments and him sensitivity about the importance of liver disease early detection.
The main reason why we included sugar consumption in coffee in the questionnaire given to the volunteers is because it is part of daily habits and is easily quantifiable, compared to the general sugar consumption in the subject’s diet.
We will certainly continue to work and spread the importance of screening for early identification of liver disease, increasing the cohort studied and collecting new data.
Comments 2: I have some minor suggestions:
In Line 36: The statement, 'The epidemic spread of SLD associates not only with an increasing incidence……' needs a change.
Suggestion: The epidemic spread of SLD is associated not only with ……
In line 37: MASLD should be Metabolic dysfunction-associated steatotic liver disease.
In 121: 'statistically differences' should be 'stastically significant differences'.
Line 137: 'oxigen' should be 'oxygen'
Response 2: Thank you for pointing out these incorrections, we fixed all of them as suggested, hoping we improved the overall value of the manuscript.
Response to Comments on the Quality of English Language
Comments: The English language use is impressive. The minor suggestions I have pointed out above should improve the overall quality.
Response: Thank you, we really hope that now the manuscript is improved and suitable to be published.

Reviewer 4 Report
Comments and Suggestions for Authors
1. For ethical purposes, I think this study requires the institutional review board statement, but it is not presented.
2. In this study, the authors should enhance the quality of the work by performing additional statistical analyses.
3. The authors have presented the information given in the tables as well as in paragraphs, but the interpretation and discussion of these values are necessary.
4. The article should be enhanced by also addressing the formulations and methods used in the study.
Author Response
Comments 1: For ethical purposes, I think this study requires the institutional review board statement, but it is not presented.
Response 1: Detailed information about IRB has been then added with the support of Assistant Editor in the section Institutional Review Board Statement (page 8, lines 305-309), as follow:
“Institutional Review Board Statement: the study was conducted in accordance with the Declaration of Helsinki; as this study is based on data collected from a voluntary, non-interventional public health screening program, Ethics Committee approval was deemed unnecessary, as per Italian law TULPS R.D. 18 June 1931, n. 773 and subsequent changes/additions, since no additional interventions were performed for this study.”
Comments 2: In this study, the authors should enhance the quality of the work by performing additional statistical analyses.
Response 2: Following the Reviewer’s suggestion, we included additional statistical analyses. In particular, we performed a more in-depth evaluation of subjects with type 2 diabetes. Type 2 diabetes represents a disease with rising prevalence in the Italian population and, more broadly, in Western countries which can significantly impact the overall population health.
Accordingly, we included the new Table 4 and the following text supporting it in the Results section (page 6, lines 206-221):
“The effect of the presence of T2D on both LS and UAP values have been also analysed and results are reported in Table 4. Both UAP and LS showed significant correlations with BMI (R=0.441, R=0.334, for UAP and LS respectively) and waist circumference (R=0.324, R=0.307, for UAP and LS, respectively) in the group without T2D (all with p<0.001). Interesting, correlations between UAP and both BMI and waist circumference become stronger in the subgroup of subjects with T2D (N=18) (R=0.553/0.468 p<0.05 for both, respectively); on the other hand, no significant correlation were found with LS. Furthermore, by comparing LS and CAP measurements in group of subjects with/without T2D stratified by sex, statistically differences were found only for LS values in Female group (p<0.05).”
Note that changes regarding new analysis performed were also reported in Statistical analysis sub-chapter of Materials and Methods (page 3, lines 113-115).
Comments 3: The authors have presented the information given in the tables as well as in paragraphs, but the interpretation and discussion of these values are necessary.
Response 3: Thank you for your comment. Accordingly, we added more details in the Results section in order to explain additional results reported in the Table (2 and 3 mainly). We also underlined the achieved results in the Discussion section. Accordingly, in the following paragraphs the two main additions were reported.
Results (page 6, lines 189-205):
“As reported in Table 2, in the univariate analysis, BMI and waist circumference values were found to have significant positive associations (p<0.001), while sex showed a significant negative relationship (p=0.004) with UAP. Other variables, including age, hypertension (HT), type 2 diabetes (T2D), alcohol consumption, smoking status, coffee consumption (sugary, non-sugary, ex-sugary), and physical activity levels, did not show significant associations individually.
In the multivariate analysis, two models were evaluated, and, in both models, BMI remained significant (p<0.001), while other variables that were significative at univariate analysis, lost their significance.
Furthermore, the univariate analysis of Table 3 shows that BMI (p<0.001) and waist circumference (p<0.001) have a significant positive associations, while sex demonstrated a significant negative association (p=0.004) with LS. All other variables, including hypertension (HT), type 2 diabetes (T2D), alcohol consumption, smoking status (never, ex, current), sugary coffee, non-sugary coffee, ex-sugary coffee drinkers, and physical activity (poor, moderate, good), showed no significant associations with stiffness variable.
In the multivariate analysis, as reported for UAP, even for LS the only variable that remained significant was BMI (p<0.05) for both models.”
Discussion (page 8, lines 262-266):
“Therefore, BMI emerged as the most consistent and significant predictor for both UAP and LS variables across models tested, while other factors, including sex and waist circumference, were significant only in univariate analysis but lost significance when adjusted for other variables.”
Comments 4: The article should be enhanced by also addressing the formulations and methods used in the study.
Response 4: Detailed information about study design, methods and other sections has been added in the text, as also requested by Reviewers 1 and 2. We hope that our manuscript in the revised form is clearly improved and suitable to be published.
